# The Changes in Various Physio-Biochemical Parameters and Yield Traits of Faba Bean Due to Humic Acid Plus 6-Benzylaminopurine Application under Deficit Irrigation

Khaled M. A. Ramadan [1,2], Hossam S. El-Beltagi [3,4,*], Taia A. Abd El-Mageed [5], Hani S. Saudy [6,*], Hala Hazam Al-Otaibi [7] and Mohamed A. A. Mahmoud [2]

1   Central Laboratories, Department of Chemistry, King Faisal University, Al-Ahsa 31982, Saudi Arabia; kramadan@kfu.edu.sa
2   Department of Agricultural Biochemistry, Faculty of Agriculture, Ain Shams University, Hadayek Shobra, Cairo 11241, Egypt; mohamed_mahmoud1@agr.asu.edu.eg
3   Agricultural Biotechnology Department, College of Agriculture and Food Sciences, King Faisal University, Al-Ahsa 31982, Saudi Arabia
4   Biochemistry Department, Faculty of Agriculture, Cairo University, Gamma St., Giza 12613, Egypt
5   Soil and Water Department, Faculty of Agriculture, Fayoum University, Fayoum 63514, Egypt; taa00@fayoum.edu.eg
6   Agronomy Department, Faculty of Agriculture, Ain Shams University, 68-Hadayek Shoubra, Cairo 11241, Egypt
7   Food and Nutrition Science Department, Agricultural Science and Food, King Faisal University, Al-Ahsa 31982, Saudi Arabia; hhalotaibi@kfu.edu.sa
*   Correspondence: helbeltagi@kfu.edu.sa (H.S.E.-B.); hani_saudy@agr.asu.edu.eg (H.S.S.)

**Abstract:** Implementing the deficit irrigation pattern has become a major strategy in crop production systems. However, using less water than is required to irrigate crops is associated with changes in plant physiology and lower productivity. Therefore, the current research aimed to assess the integrated effect of humic acid and cytokinin on faba bean under water deficit. Under two irrigation levels (full irrigation, FI and deficit irrigation, DI), two humic acid treatments (without addition, $H_0$ and with addition of 10 kg ha$^{-1}$, $H_{10}$) and two cytokinin concentrations (without spray, $C_0$ and spraying with 25 mg L$^{-1}$, $C_{25}$), faba bean growth, physiology, and productivity were evaluated. The experiment was implemented for two winter seasons of 2019/20 and 2020/21 and performed in a split–split plots design with three replicates. The findings revealed that under low water supply (DI), $H_{10}$ plus $C_{25}$ was the most efficient treatment for enhancing faba bean growth. All physiological faba bean traits estimated under DI showed remarkable increases with the application of $H_{10}$ plus $C_{25}$ in both seasons. The increases in proline, catalase, and total soluble sugars under DI due to $H_{10}$ plus $C_{25}$ were 31.4 and 31.8%, 51.9 and 55.1% as well as 43.8 and 46.6%, in the first and second seasons, respectively. There was no significant difference between FI × $H_{10}$ plus $C_{25}$ and DI × $H_{10}$ plus $C_{25}$ in phosphorus content in both seasons. FI × $H_{10}$ plus $C_{25}$ and DI × $H_{10}$ plus $C_{25}$ in the second season produced a similar number of pods plant$^{-1}$ and seed yield of faba bean. Conclusively, the combined application of humic plus cytokinin achieved physiological and nutrient homeostasis, adjusting the biochemical compounds in faba bean under water deficit.

**Keywords:** drought stress; faba bean yield; osmo-protectants; physiological homeostasis; seed nutrient contents; water use efficiency; chemometrics

## 1. Introduction

Faba bean (*Vicia faba* L.), as a member of the Fabaceae family, has seeds rich in protein, minerals, and vitamins [1]. Unfortunately, the yield obtained from stressed faba bean plants had undesirable properties both in terms of quantity and quality [2–4].

It is well documented that drought causes changes in plant physiology [5] and biochemical constituents [6,7]. Furthermore, water deficit disrupts nutrient homeostasis in plants. Thus, lower crop yield and quality are obtained under drought conditions [8]. Several physiological and biochemical indices are associated with drought tolerance in plants. Plants respond and become acclimatized to drought stress by modulating many physiological, biochemical, and molecular aspects [9]. Under drought, the metabolic activity in plant cells is influenced by the relative water content (RWC), which decreases in drought-affected plant tissues [10]. Further, the membrane stability index (MSI) is a physiological indicator for drought tolerance, since a reduction in cell membrane stability refers to reactive oxygen species (ROS)-generated oxidation of lipid peroxidation [11]. Furthermore, photo-oxidation and disintegration of chlorophyll, expressed in the chlorophyll stability index (CSI), are features reflect strongly drought-affected plant status, correlating with crop yield [12]. Additionally, drought tolerance in plants is positively correlated with maintaining a high level of enzymatic and non-enzymatic antioxidants [13–15]. Herein, proline as a non-enzymatic antioxidant and catalase as an enzymatic antioxidant can scavenge and/or suppress the production of ROS in plant organelles under oxidative stresses [10,16].

Humic acid is involved in numerous organic complexes and has various active chemical groups [17,18]. In these compounds, the abundance of humic acid improves the availability of nutrients in soil and mineral uptake by plants [19–22]. Applications of humic acid serve the plant via increasing root growth, stimulating soil microorganisms, increasing water holding, or soil aggregation [19,23]. Consequently, humic acid-treated plants had better root growth, hence productivity, than non-treated plants [17,24]. Further, drought can be effectively overcome by the exogenous application of plant growth regulators to motivate plant tolerance to various abiotic stresses [25]. The increases associated with the hormonal products for plant tolerance to various stresses can be attributed to stimulating plants' detoxifying potential and adjusting physiological behavior [26,27].

Furthermore, growth regulators can mitigate the adverse impacts of drought by increasing and upregulating antioxidant-based enzymes and osmo-protectants, reducing the peroxidation of lipids [28,29]. Cytokinins, as distinctive growth regulators, have diverse roles in plant development, involving cell growth and differentiation [30]. Further, reports have alluded the significance of cytokinins as moderators of cellular readjustment responses to drought [31,32]. Cytokinin compound application alleviated osmotic stress by delaying leaf senescence and reducing physiological deterioration [33,34]. There is copious evidence showing that cytokinins assist in better plant growth under osmotic stress conditions, eventually leading to improvements in crop yield [35–37]. Despite the clear role of humic substances and growth regulators on plant growth and development, the interactive effect of humic acid and cytokinin on faba bean under water deficiency requires further investigation.

In this work, we hypothesize that humic acid plus cytokinin can increase physiological balance and improve the quantity and quality of faba bean seeds. Therefore, this study aimed to assess the changes in growth, physiological status, biochemical compounds, nutrient content and yield traits of faba bean due to humic acid and cytokinin interaction under full and deficit irrigation.

## 2. Materials and Methods

### 2.1. Experimental Site Description

At a private farm in the El Fayoum region of Egypt (latitudes 29°06′ and 29°35′ N, longitudes 30°26′ and 31°05′ E, and altitude: −3 m.a.s.l.), field trials were conducted over two succeeding seasons (2019/20 and 2020/21). Additionally, the soil's primary physio-chemical properties were assessed in accordance with Klute and Dirksen and Page [38,39]. The soil is a loamy sand texture containing sand (75.4%), silt (12.5%), and clay (12.1%), with a bulk density of 1.54 g cm$^{-3}$, a pH of 7.66, an electrical conductivity of saturation extract, ECe, of 5.24 dSm$^{-1}$, and a cation exchange capacity of 12.3 cmol kg$^{-1}$, as well as the following amounts of nutrients: calcium carbonate (4.2%), organic carbon (1.06%), available N,

(57.2 mg kg$^{-1}$ soil), available P (4.4 mg kg$^{-1}$ soil), available K (52.1 mg kg$^{-1}$ soil) and available Zn (0.78 mg kg$^{-1}$ soil). The experimental site was located in an arid region with moderate winters and rare precipitation.

### 2.2. Agronomic Management and Treatments

Faba bean healthy seeds (*Vicia faba*. L., cultivar Sakha 1) were sown on October 15 and 20 in 2019 and 2020 and harvested on April 21 and 27 in 2020 and 2021, respectively. Treatments involved the combination of irrigation levels, humic acid, and cytokinin spray. Two irrigation levels based on crop evapotranspiration (Etc), full irrigation (FI, 100% of Etc), and deficit irrigation (DI, 80% of Etc) were applied. There were two rates of humic acid (without H$_0$ and with the application of 10 kg ha$^{-1}$, H$_{10}$) as well as foliar spray with synthetic cytokinin, 6-benzylaminopurine (without spraying, plants were sprayed with distilled water C$_0$, and spraying with 25 mg L$^{-1}$, C$_{25}$). Plants were treated with cytokinin twice at 30 and 45 days after sowing. Humic acid was added once during planting and it was mixed well with the appropriate amount of sand (~200 kg), and then evenly distributed over the top layer of the soil and mixed in the rhizosphere zone. Irrigation levels were allocated in main plots, while humic acid was distributed in the sub-plots. Finally, the cytokinin levels were decreased in the sub-sub-plots. A total of nine treatments were replicated three times via a randomized complete split–split plot block design, resulting in a total of 24 experimental plots. The experimental plots were 12.8 m$^2$ in size (0.8 × 16 m), with two planting rows; the rows were 1 m in width, with 15 cm spacing between each plant. A drip irrigation system was utilized, and 2 drip lines were placed 30 cm apart in every elementary test plot. Irrigation treatments began after the full germination stage. Phosphorus (P) and potassium (K) fertilizers were added at planting at a rate of 75 kg P ha$^{-1}$ in the form of calcium superphosphate (15.5% P$_2$O$_5$) and 120 kg K ha$^{-1}$ in the form of potassium sulfate (48% K$_2$O), respectively. Nitrogen (N) fertilizer was added once as a starter dose at planting at a rate of 48 kg N ha$^{-1}$ in the form of ammonium nitrate (33.5% N).

### 2.3. Irrigation Water Applied

According to the FAO Penman–Monteith equation, the daily reference evapotranspiration (Eto) was calculated using the following formula [39]:

$$Etc = Eto \times Kc \tag{1}$$

where Etc is the crop water requirement (mm d$^{-1}$) and Kc is the crop coefficient.

The irrigation water applied (IWA) per bed was calculated according to the following equation:

$$IWA = \frac{Etc \times A \times Ii}{Ea \times 1000} \tag{2}$$

where IWA is the irrigation water applied (m$^3$), A is the plot area (m$^2$), Ii is the irrigation period (day), and Ea is the irrigation efficiency (%).

### 2.4. Measurements

#### 2.4.1. Water Status and Photosynthetic Capacity

At 75 days after sowing, the relative water content (RWC%) and the membrane stability index (MSI%) were assessed [40,41]. To assess the photosynthetic efficiency, the performance index, and chlorophyll fluorescence were determined according to Clark et al. [42] and Maxwell and Johnson [43] by Handy PEA, Hansatech Instruments (Ltd., Kings Lynn, London, UK). Additionally, leaf greenness (SPAD value) was determined using a chlorophyll meter (SPAD502, KONICAMINOLTA. Inc., Tokyo, Japan).

### 2.4.2. Free Proline Content, Total Soluble Sugars and Enzyme

The free proline content and total soluble sugars (TSS) (mg g$^{-1}$ FW) of fresh faba bean leaves were extracted and quantified utilizing procedures described previously [44,45]. Plant cells were extracted following the technique of Bradford [46] for use as a crude enzyme extract to measure CAT content. The CAT activity (EC 1.11.1.6) was established using the approach published by Aebi (Burgdorf, Switzerland) [47].

### 2.4.3. Growth Traits

At the end of the growing season, ten plants were randomly obtained from every experimental plot and assessed for their growth characteristics. Plant height was recorded as well as the number of leaves and branches plant$^{-1}$. Total leaf area plant$^{-1}$, was measured using a digital plan meter, Planix 7 (Sokkia Co., Ltd. Atsugi, Kanagawa, Japan). Shoot dry weight plant$^{-1}$ was recorded after oven-drying at 70 °C until constant weight.

### 2.4.4. Leaf Mineral Contents

To assess the contents of N, P, and K, faba bean leaves were dried and grounded to form a powder. The digestion process was performed for the dried samples with a mixture consisting of $HClO_4$ and $H_2SO_4$ (at 1:3 *v/v*, respectively). N content was assessed using micro-Kjeldahl equipment (Ningbo Medical Instruments Co., Ningbo, China [48]. Molybdenum blue, diluted $H_2MoO_7S$, and 8% (*w/v*) $NaHSO_3$-$H_2SO_4$ were used as standard reagents for quantifying P [49]. K contents were measured using a Perkin-Elmer Model 52-A Flame Photometer (Waltham, MA, USA) Jackson [50].

### 2.4.5. Yield and Yield Components

At harvesting stage, 10 plants were randomly selected from each plot and utilized to determine yield components, i.e., the number of pods per plant and 100-seed weight. Seeds of all plants per plot were utilized to determine seed yield (t ha$^{-1}$).

### 2.4.6. Water Use Efficiency

According to Fernández et al. [51], water use efficiency (WUE) was computed using the formula given below:

$$\text{WUE} = \frac{\text{Seed yield}\left(\text{kg ha}^{-1}\right)}{\text{Water applied}\left(\text{m}^3\text{ ha}^{-1}\right)} \tag{3}$$

### 2.5. Statistical Analysis

Data were statistically evaluated following Gomez and Gomez [52] with analysis of variance procedures in the GenStat statistical package (version 11) (VSN International Ltd., Oxford, UK). Data for each growing season were subjected to two-way analysis of variance (ANOVA). The Duncan multiple range test, at a 0.05 probability level, was utilized to compare treatment means. Further, data preparation for chemometric analysis was according to Mahmoud et al. and Mahmoud and Magdy [53,54]. Agglomerative hierarchical clustering (AHC) and principal component analysis (PCA) were used in XLSTAT 2022® (Addinsoft, Paris, France).

## 3. Results

### 3.1. Growth Response

Growth of faba bean significantly responded to the combinations of humic acid and cytokinin in the 2019/20 and 2020/21 seasons (Table 1). The maximum values for all growth traits were more pronounced with FI × H$_{10}$ plus C$_{25}$, statistically equal in FI × H$_{10}$ plus C$_0$ and FI × H$_0$ plus C$_{25}$ as well as DI × H$_{10}$ plus C$_{25}$ and DI × H$_0$ plus C$_{25}$ in number of branches plant$^{-1}$ in the first season. Under FI, the combinations of H$_{10}$ plus C$_{25}$ (for all traits), H$_{10}$ plus C$_0$ (for number of branches plant$^{-1}$ and dry matter plant$^{-1}$)



and $H_0$ plus $C_{25}$ (for leaf area and dry matter plant$^{-1}$) exhibited the maximum values in the second season. Furthermore, under DI, $H_{10}$ plus $C_{25}$ was the most efficient treatment for enhancing faba bean growth, significantly similar to $H_{10}$ plus $C_0$ in all growth traits, in both seasons, except plant height and leaf area in the first season. Compared to the counterpart treatment (DI × $H_0$ plus $C_0$ $H_0$), the DI × $H_{10}$ plus $C_{25}$ treatment increased plant height (by 22.3 and 23.5%), number of leaves plant$^{-1}$ (by 21.2 and 12.8%), number of branches plant$^{-r}$ (by 50.0 and 35.1%), leaf area (by 22.2 and 23.6%) and dry matter plant$^{-n}$ (by 22.4 and 23.5%) in the first and second seasons, respectively.

**Table 1.** Faba bean growth as influenced by humic acid plus cytokinin treatments under irrigation regimes in the 2019/2020 and 2020/2021 seasons.

| Season | Irrigation Regime | Treatments | | Plant Height (cm) | Number of Leaves Plant$^{-1}$ | Number of Branches Plant$^{-1}$ | Leaf Area (dm$^2$) | Dry Matter Plant$^{-1}$ (g) |
|---|---|---|---|---|---|---|---|---|
| 2019/2020 | FI | $H_0$ | $C_0$ | 87.1 ± 0.88 [d] | 97.0 ± 1.16 [d] | 5.56 ± 0.11 [b] | 189.7 ± 0.94 [g] | 49.7 ± 0.81 [f] |
| | | | $C_{25}$ | 90.7 ± 0.33 [c] | 101.3 ± 0.88 [c] | 5.89 ± 0.22 [ab] | 233.3 ± 1.3 [b] | 56.9 ± 1.1 [bc] |
| | | $H_{10}$ | $C_0$ | 95.7 ± 0.33 [b] | 106.6 ± 0.37 [b] | 6.33 ± 0.33 [a] | 227.6 ± 0.79 [c] | 58.0 ± 0.20 [b] |
| | | | $C_{25}$ | 103.0 ± 0.58 [a] | 114.9 ± 0.64 [a] | 6.00 ± 0.11 [ab] | 245.0 ± 1.37 [a] | 62.5 ± 0.35 [a] |
| | DI | $H_0$ | $C_0$ | 75.2 ± 0.67 [e] | 73.7 ± 2.6 [g] | 4.00 ± 0.00 [c] | 178.6 ± 1.58 [h] | 45.61 ± 0.40 [g] |
| | | | $C_{25}$ | 86.6 ± 0.67 [d] | 83.3 ± 1.8 [f] | 6.00 ± 0.33 [ab] | 205.5 ± 1.58 [f] | 52.3 ± 0.40 [e] |
| | | $H_{10}$ | $C_0$ | 90.3 ± 0.33 [c] | 87.6 ± 0.32 [e] | 5.67 ± 0.32 [b] | 214.4 ± 0.79 [e] | 54.8 ± 0.20 [d] |
| | | | $C_{25}$ | 92.0 ± 0.58 [c] | 89.3 ± 0.56 [e] | 6.00 ± 0.33 [ab] | 218.3 ± 1.37 [d] | 55.8 ± 0.35 [cd] |
| 2020/2021 | FI | $H_0$ | $C_0$ | 84.5 ± 2.02 [de] | 90.3 ± 3.38 [d] | 4.3 ± 0.33 [c] | 184.1 ± 2.65 [ef] | 48.2 ± 0.92 [d] |
| | | | $C_{25}$ | 94.7 ± 4.33 [bc] | 93.8 ± 0.87 [c] | 4.6 ± 0.32 [bc] | 243.3 ± 10.3 [ab] | 59.5 ± 3.5 [ab] |
| | | $H_{10}$ | $C_0$ | 96.7 ± 0.33 [b] | 108.8 ± 0.32 [b] | 5.6 ± 0.33 [a] | 230.0 ± 0.97 [bc] | 58.6 ± 0.20 [abc] |
| | | | $C_{25}$ | 104.0 ± 0.53 [a] | 116.9 ± 0.56 [a] | 6.0 ± 0.31 [a] | 247.4 ± 1.37 [a] | 63.1 ± 0.5 [a] |
| | DI | $H_0$ | $C_0$ | 73.7 ± 0.88 [f] | 78.3 ± 0.33 [e] | 3.7 ± 0.34 [d] | 174.8 ± 2.1 [f] | 44.7 ± 0.53 [d] |
| | | | $C_{25}$ | 80.3 ± 2.7 [e] | 83.5 ± 6.7 [de] | 4.3 ± 0.33 [c] | 190.7 ± 6.3 [e] | 48.7 ± 1.6 [d] |
| | | $H_{10}$ | $C_0$ | 89.3 ± 0.33 [cd] | 86.7 ± 0.37 [cde] | 4.6 ± 0.29 [bc] | 212.0 ± 0.97 [d] | 54.2 ± 0.20 [c] |
| | | | $C_{25}$ | 91.0 ± 0.58 [bc] | 88.3 ± 0.64 [cd] | 5.0 ± 0.33 [b] | 216.0 ± 1.4 [cd] | 55.2 ± 0.35 [bc] |

Each value indicates the mean ± standard error (*n* = 3). Mean values in each column followed by the same lower-case letter in each column are not significantly different according to the Duncan test ($p \leq 0.05$). FI, full irrigation; DI, deficit irrigation (80% of crop evapotranspiration); $H_0$ and $H_{10}$: without and with the application of 10 kg ha$^{-1}$ of humic acid, respectively. $C_0$ and $C_{25}$: without and with 25 mg L$^{-1}$ of cytokinin, respectively.

*3.2. Physiological Response*

The physiological changes in faba bean due to humic acid plus cytokinin under irrigation regimes are presented in Table 2. Under FI or DI, $H_{10}$ plus $C_{25}$ or $C_0$ in both seasons, in addition to FI × $H_0$ plus $C_{25}$ (for Fv/Fm in the second season), resulted in the maximum increases in SPAD and Fv/Fm (except DI × $H_{10}$ plus $C_0$ for SPAD in the first season). Moreover, FI × $H_{10}$ plus $C_{25}$ was the effective practice for improving the performance index, the relative water content, and the membrane stability index, significantly similar to $H_{10}$ plus $C_0$ for the relative water content in both seasons. It should be noted that all physiological parameters of faba bean measured under DI showed distinctive improvements with the application of $H_{10}$ plus $C_{25}$ in both seasons. Herein, under DI, $H_{10}$ plus $C_{25}$ increased SPAD, Fv/Fm, the performance index, the relative water content and the membrane stability index by approximately 1.50 and 1.50, 1.10 and 1.10, 3.76 and 3.32, 1.18 and 1.17 and 1.32 and 1.6 fold, in the first and second seasons, respectively, compared to $H_0$ plus $C_0$.

*3.3. Biochemical Compounds*

Humic plus cytokinin had a significant effect on proline (Figure 1), catalase (Figure 2), and total soluble sugars (Figure 3) in both seasons of 2019/20 and 2020/21. FI × $H_{10}$ plus $C_{25}$ resulted in the maximum value of proline, surpassing that of FI × $H_0$ plus $C_0$ by 47.9 and 48.4% in the first and second seasons, respectively. Moreover, $H_{10}$ plus $C_{25}$ resulted

in the highest values of catalase and total soluble sugars whether with FI or DI in both seasons, except catalase under DI in the second season. The increases in proline, catalase, and total soluble sugars under DI due to $H_{10}$ plus $C_{25}$ amounted to 31.4 and 31.8%, 51.9 and 55.1%, as well as 43.8 and 46.6%, in 2019/20 and 2020/21, respectively.

**Table 2.** Physiological response of faba bean as influenced by humic acid plus cytokinin treatments under irrigation regimes in the 2019/2020 and 2020/2021 seasons.

| Season | Irrigation Regime | Treatments | | SPAD | Fv/Fm | Performance Index | Relative Water Content | Membrane Stability Index |
|---|---|---|---|---|---|---|---|---|
| 2019/2020 | FI | $H_0$ | $C_0$ | 44.6 ± 0.80 [b] | 0.79 ± 0.01 [c] | 3.1 ± 0.02 [f] | 78.34 ± 1.7 [c] | 39.0 ± 1.3 [d] |
| | | | $C_{25}$ | 46.0 ± 1.4 [b] | 0.82 ± 0.00 [b] | 5.0 ± 0.12 [de] | 80.7 ± 0.66 [bc] | 55.1 ± 1.6 [b] |
| | | $H_{10}$ | $C_0$ | 50.7 ± 0.60 [a] | 0.83 ± 0.03 [ab] | 6.2 ± 0.87 [d] | 82.8 ± 0.34 [ab] | 54.7 ± 1.8 [b] |
| | | | $C_{25}$ | 51.5 ± 0.23 [a] | 0.84 ± 0.01 [a] | 12.2 ± 0.33 [a] | 84.1 ± 0.55 [a] | 61.9 ± 1.6 [a] |
| | DI | $H_0$ | $C_0$ | 33.8 ± 1.1 [d] | 0.76 ± 0.01 [d] | 2.6 ± 0.06 [f] | 67.1 ± 0.43 [e] | 34.0 ± 1.1 [f] |
| | | | $C_{25}$ | 40.0 ± 0.573 [c] | 0.78 ± 0.00 [c] | 4.8 ± 0.48 [e] | 74.8 ± 1.2 [d] | 37.7 ±0.54 [e] |
| | | $H_{10}$ | $C_0$ | 46.2 ± 0.53 [b] | 0.83 ± 0.01 [ab] | 8.2 ± 1.2 [c] | 75.5 ± 0.79 [d] | 41.3 ± 0.88 [d] |
| | | | $C_{25}$ | 50.8 ± 0.33 [a] | 0.84 ± 0.01 [a] | 9.8 ± 0.78 [b] | 79.3 ± 0.54 [c] | 45.0 ± 0.58 [c] |
| 2020/2021 | FI | $H_0$ | $C_0$ | 43.4 ± 0.40 [d] | 0.80 ± 0.01 [b] | 2.9 ± 0.06 [f] | 79.6 ± 1.6 [c] | 36.0 ± 0.89 [f] |
| | | | $C_{25}$ | 45.9 ± 1.3 [cd] | 0.82 ± 0.01 [ab] | 4.9 ± 0.50 [e] | 81.6 ± 1.3 [bc] | 57.1 ± 1.1 [b] |
| | | $H_{10}$ | $C_0$ | 50.5 ± 0.81 [ab] | 0.82 ± 0.01 [ab] | 5.9 ± 0.33 [de] | 83.7 ± 0.79 [ab] | 56.7 ± 0.88 [b] |
| | | | $C_{25}$ | 52.1 ± 0.57 [a] | 0.83 ± 0.01 [a] | 12.8 ± 0.58 [a] | 85.1 ± 1.5 [a] | 63.9 ± 0.58 [a] |
| | DI | $H_0$ | $C_0$ | 33.3 ± 1.3 [e] | 0.76 ± 0.01 [d] | 2.8 ± 0.20 [f] | 68.2 ± 1.4 [e] | 29.3 ± 0.33 [g] |
| | | | $C_{25}$ | 48.5 ± 1.2 [bc] | 0.79 ± 0.01 [c] | 6.3 ± 0.80 [cd] | 74.7 ± 1.2 [d] | 39.7 ± 0.33 [e] |
| | | $H_{10}$ | $C_0$ | 51.6 ± 1.1 [a] | 0.84 ± 0.01 [a] | 7.6 ± 0.25 [c] | 77.3 ± 1.7 [d] | 43.3 ± 0.86 [d] |
| | | | $C_{25}$ | 50.1 ± 0.55 [ab] | 0.84 ± 0.01 [a] | 9.3 ± 0.82 [b] | 80.0 ± 0.54 [c] | 47.0 ± 1.1 [c] |

Each value indicates the mean ± standard error (*n* = 3). Mean values in each column followed by the same lower-case letter in each column are not significantly different according to the Duncan test ($p \leq 0.05$). FI, full irrigation; DI, deficit irrigation (80% of crop evapotranspiration); $H_0$ and $H_{10}$: without and with the application of 10 kg ha$^{-1}$ of humic acid, respectively. $C_0$ and $C_{25}$: without and with 25 mg L$^{-1}$ of cytokinin, respectively.

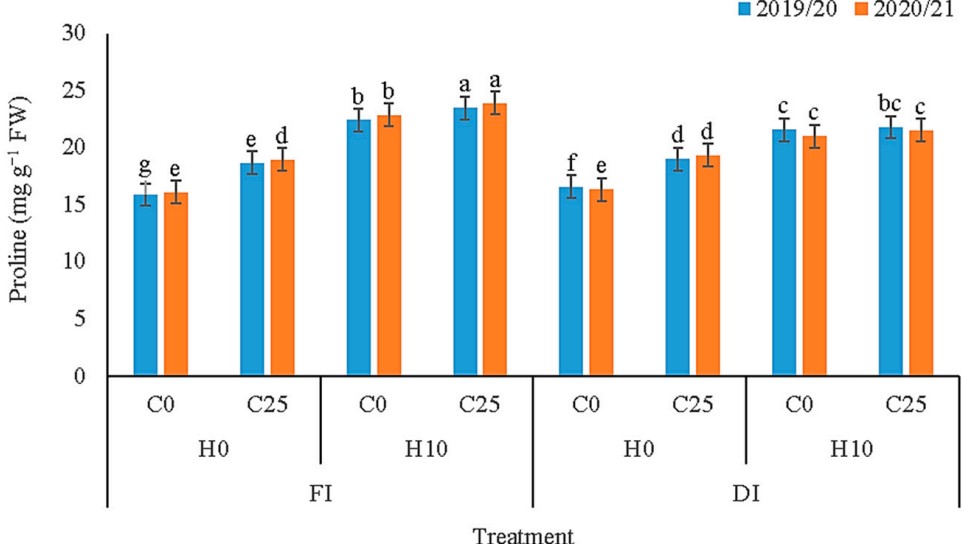

**Figure 1.** Proline content of faba bean as influenced by humic acid plus cytokinin treatments under irrigation regimes in the 2019/20 and 2020/21 seasons. Each value indicates the mean ± standard error (*n* = 3). Mean values in each bar followed by the same letter are not significantly different according to the Duncan test ($p \leq 0.05$). FI, full irrigation; DI, deficit in irrigation (80% of crop evapotranspiration); $H_0$ and $H_{10}$: without and with the application of 10 kg ha$^{-1}$ of humic acid, respectively. $C_0$ and $C_{25}$: without and with 25 mg L$^{-1}$ of cytokinin, respectively.

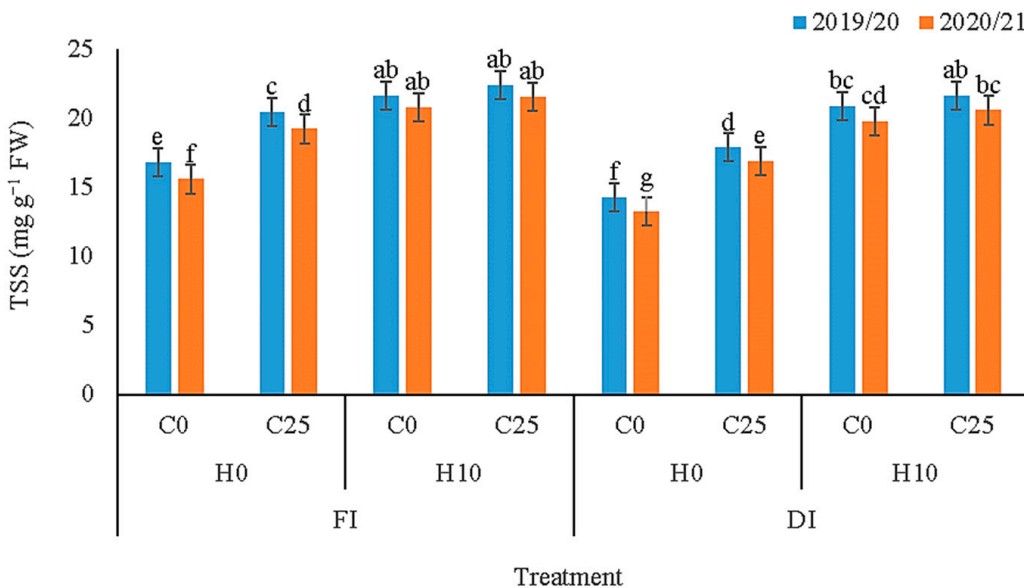

**Figure 2.** Total soluble sugars (TSS) content of faba bean as influenced by humic acid plus cytokinin treatments under irrigation regimes in the 2019/20 and 2020/21 seasons. Each value indicates the mean $\pm$ standard error ($n$ = 3). Mean values in each bar followed by the same letter are not significantly different according to the Duncan test ($p \leq 0.05$). FI, full irrigation; DI, deficit in irrigation (80% of crop evapotranspiration); $H_0$ and $H_{10}$: without and with the application of 10 kg ha$^{-1}$ of humic acid, respectively. $C_0$ and $C_{25}$: without and with 25 mg L$^{-1}$ of cytokinin, respectively.

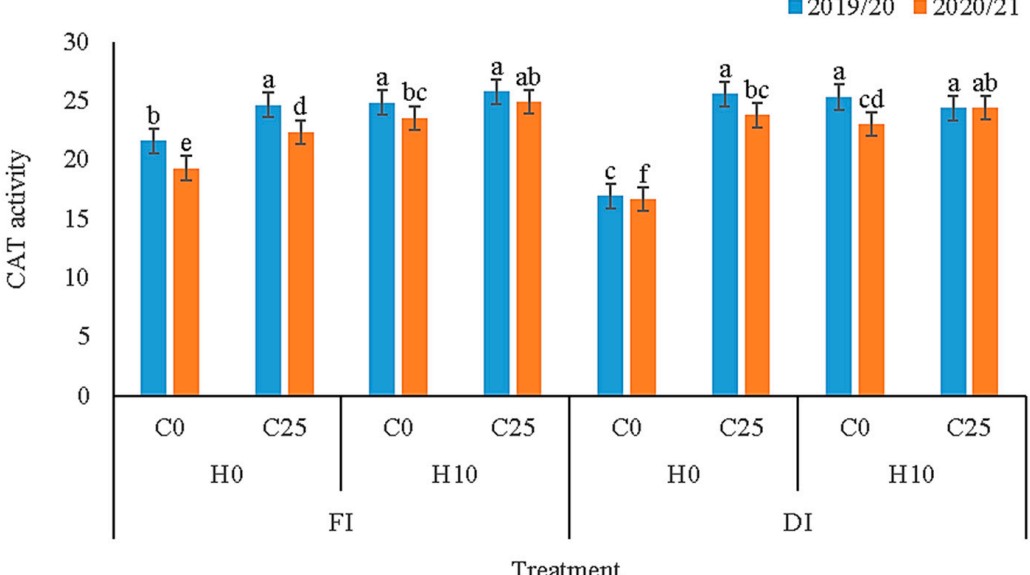

**Figure 3.** Catalase (CAT) activity of faba bean as influenced by humic acid plus cytokinin treatments under irrigation regimes in the 2019/20 and 2020/21 seasons. Each value indicates the mean $\pm$ standard error ($n$ = 3). Mean values in each bar followed by the same letter are not significantly different according to the Duncan test ($p \leq 0.05$). FI, full irrigation; DI, deficit in irrigation (80% of crop evapotranspiration); $H_0$ and $H_{10}$: without and with the application of 10 kg ha$^{-1}$ of humic acid, respectively. $C_0$ and $C_{25}$: without and with 25 mg L$^{-1}$ of cytokinin, respectively.

### 3.4. Nutrient Contents

As shown in Table 3, the nutrient content of faba bean markedly changed based on the combinations of humic acid and cytokinin in the 2019/20 and 2020/21 seasons. In this respect, the application of FI $\times$ $H_{10}$ plus $C_{25}$ resulted in the highest values of nitrogen,

phosphorus and potassium in both seasons. However, the difference between FI $\times$ $H_{10}$ plus $C_{25}$ and DF $\times$ $H_{10}$ plus $C_{25}$ in terms of phosphorus content was not significant in both seasons. Compared to their counterpart control treatments, the increases in phosphorus content due to $H_{10}$ plus $C_{25}$ application under FI and DI was 1.52 and 2.24 fold in the first season and 1.46 and 2.17 fold in the second season, respectively.

**Table 3.** Leaf nutrient contents of faba bean as influenced by humic acid plus cytokinin treatments under irrigation regimes in the 2019/2020 and 2020/2021 seasons.

| Season | Irrigation Regime | Treatments | | Nitrogen mg/g DW | Phosphorus mg/g DW | Potassium mg/g DW |
|---|---|---|---|---|---|---|
| 2019/2020 | FI | $H_0$ | $C_0$ | 16.59 $\pm$ 0.30 [d] | 3.87 $\pm$ 0.10 [e] | 14.47 $\pm$ 0.46 [c] |
| | | | $C_{25}$ | 19.52 $\pm$ 0.24 [b] | 5.19 $\pm$ 0.16 [bc] | 17.74 $\pm$ 0.50 [b] |
| | | $H_{10}$ | $C_0$ | 18.36 $\pm$ 0.25 [c] | 4.87 $\pm$ 0.06 [cd] | 15.79 $\pm$ 0.45 [c] |
| | | | $C_{25}$ | 21.49 $\pm$ 0.24 [a] | 5.87 $\pm$ 0.15 [a] | 20.15 $\pm$ 0.51 [a] |
| | DI | $H_0$ | $C_0$ | 9.82 $\pm$ 0.59 [g] | 2.50 $\pm$ 0.29 [g] | 12.46 $\pm$ 0.92 [d] |
| | | | $C_{25}$ | 15.15 $\pm$ 0.53 [e] | 4.60 $\pm$ 0.31 [d] | 15.67 $\pm$ 0.33 [c] |
| | | $H_{10}$ | $C_0$ | 11.92 $\pm$ 0.59 [f] | 3.28 $\pm$ 0.28 [f] | 14.12 $\pm$ 0.60 [cd] |
| | | | $C_{25}$ | 17.49 $\pm$ 0.52 [cd] | 5.60 $\pm$ 0.30 [ab] | 17.76 $\pm$ 0.33 [b] |
| 2020/2021 | FI | $H_0$ | $C_0$ | 17.69 $\pm$ 0.30 [c] | 4.11 $\pm$ 0.06 [d] | 15.56 $\pm$ 0.45 [de] |
| | | | $C_{25}$ | 20.51 $\pm$ 0.23 [b] | 5.51 $\pm$ 0.15 [ab] | 18.83 $\pm$ 0.51 [bc] |
| | | $H_{10}$ | $C_0$ | 19.66 $\pm$ 0.20 [b] | 5.01 $\pm$ 0.10 [bc] | 17.09 $\pm$ 0.46 [cd] |
| | | | $C_{25}$ | 21.88 $\pm$ 0.20 [a] | 6.03 $\pm$ 0.16 [a] | 21.45 $\pm$ 0.50 [a] |
| | DI | $H_0$ | $C_0$ | 10.81 $\pm$ 0.58 [f] | 2.67 $\pm$ 0.28 [f] | 13.96 $\pm$ 0.91 [e] |
| | | | $C_{25}$ | 16.14 $\pm$ 0.52 [d] | 4.73 $\pm$ 0.30 [c] | 17.17 $\pm$ 0.30 [cd] |
| | | $H_{10}$ | $C_0$ | 12.81 $\pm$ 0.58 [e] | 3.43 $\pm$ 0.30 [e] | 15.77 $\pm$ 0.60 [de] |
| | | | $C_{25}$ | 18.38 $\pm$ 0.53 [c] | 5.80 $\pm$ 0.30 [a] | 19.41 $\pm$ 0.30 [b] |

Each value indicates the mean $\pm$ standard error ($n$ = 3). Mean values in each column followed by the same lower-case letter in each column are not significantly different according to the Duncan test ($p \leq 0.05$). FI, full irrigation; DI, deficit irrigation (80% of crop evapotranspiration); $H_0$ and $H_{10}$: without and with the application of 10 kg $ha^{-1}$ of humic acid, respectively. $C_0$ and $C_{25}$: without and with 25 mg $L^{-1}$ of cytokinin, respectively.

*3.5. Yield Traits and Water Use Efficiency*

The number of pods $plant^{-1}$, the weight of 100 seeds and the seed yield of faba bean showed significant changes in response to humic acid and cytokinin applications in the 2019/20 and 2020/21 seasons (Table 4). The most effective practice for increasing all yield traits in both seasons was the application of $H_{10}$ plus $C_{25}$ under FI. In the first season, FI $\times$ $H_0$ plus $C_{25}$ showed similar values for the number of pods $plant^{-1}$ and DI $\times$ $H_{10}$ plus $C_{25}$ showed similar values for seed yield to that of FI $\times$ $H_{10}$ plus $C_{25}$. FI $\times$ $H_{10}$ plus $C_{25}$ and DI $\times$ $H_{10}$ plus $C_{25}$ produced a similar number of pods $plant^{-1}$ and seed yield in the second season. It must be pointed out that the application of $H_{10}$ plus $C_{25}$ under DI achieved resulted in increases of 30.8 and 46.7% in the number of pods $plant^{-1}$, 19.8 and 17.2% in the weight of 100 seeds and 20.1 and 23.1% in the seed yield compared to $H_0$ plus $C_0$ in the first and second seasons, respectively. Concerning water use efficiency (WUE), Figure 4 shows that the addition of $H_{10}$ either with $C_0$ or $C_{25}$ under DI resulted in the maximum values of WUE in both seasons, surpassing that of the other treatments. FI $\times$ $H_0$ plus $C_0$ was the least effective practice, resulting in the lowest values of WUE in both seasons.

**Table 4.** Yield parameters of faba bean as influenced by humic acid plus cytokinin treatments under irrigation regimes in the 2019/2020 and 2020/2021 seasons.

| Season | Irrigation Regime | Treatments | | Number of Pods Plant$^{-1}$ | Weight of 100 Seeds (g) | Seed Yield (t ha$^{-1}$) |
|---|---|---|---|---|---|---|
| 2019/2020 | FI | $H_0$ | $C_0$ | 12.3 ± 0.67 [de] | 80.7 ± 0.47 [c] | 4.10 ± 0.08 [ef] |
| | | | $C_{25}$ | 17.6 ± 0.89 [ab] | 87.6 ± 2.50 [b] | 4.40 ± 0.03 [cd] |
| | | $H_{10}$ | $C_0$ | 15.7 ± 0.67 [bc] | 89.7 ± 0.82 [b] | 4.53 ± 0.04 [bc] |
| | | | $C_{25}$ | 18.3 ± 0.67 [a] | 94.6 ± 0.42 [a] | 4.78 ± 0.02 [a] |
| | DI | $H_0$ | $C_0$ | 12.0 ± 0.51 [e] | 73.7 ± 0.64 [d] | 3.98 ± 0.02 [f] |
| | | | $C_{25}$ | 14.9 ± 0.48 [c] | 79.0 ± 0.60 [c] | 4.27 ± 0.03 [de] |
| | | $H_{10}$ | $C_0$ | 14.3 ± 0.33 [cd] | 86.7 ± 0.74 [b] | 4.53 ± 0.04 [bc] |
| | | | $C_{25}$ | 15.7 ± 0.67 [bc] | 88.3 ± 2.33 [b] | 4.78 ± 0.13 [a] |
| 2020/2021 | FI | $H_0$ | $C_0$ | 13.7 ± 1.33 [cd] | 82.0 ± 0.50 [d] | 4.05 ± 0.13 [d] |
| | | | $C_{25}$ | 17.0 ± 0.67 [ab] | 90.6 ± 2.70 [b] | 4.33 ± 0.07 [b] |
| | | $H_{10}$ | $C_0$ | 17.3 ± 0.58 [ab] | 87.0 ± 0.50 [bc] | 4.57 ± 0.09 [ab] |
| | | | $C_{25}$ | 19.7 ± 0.88 [a] | 96.8 ± 0.77 [a] | 4.67 ± 0.07 [a] |
| | DI | $H_0$ | $C_0$ | 12.0 ± 1.00 [d] | 76.2 ± 0.61 [e] | 3.76 ± 0.03 [e] |
| | | | $C_{25}$ | 15.0 ± 1.00 [bc] | 84.3 ± 1.67 [cd] | 4.13 ± 0.03 [cd] |
| | | $H_{10}$ | $C_0$ | 16.6 ± 0.35 [b] | 87.0 ± 0.58 [bc] | 4.57 ± 0.07 [ab] |
| | | | $C_{25}$ | 17.6 ± 0.74 [ab] | 89.3 ± 0.33 [b] | 4.63 ± 0.09 [a] |

Each value indicates the mean ± standard error (*n* = 3). Mean values in each column followed by the same lower-case letter in each column are not significantly different according to the Duncan test ($p \leq 0.05$). FI, full irrigation; DI, deficit irrigation (80% of crop evapotranspiration); $H_0$ and $H_{10}$: without and with the application of 10 kg ha$^{-1}$ of humic acid, respectively. $C_0$ and $C_{25}$: without and with 25 mg L$^{-1}$ of cytokinin, respectively.

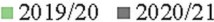

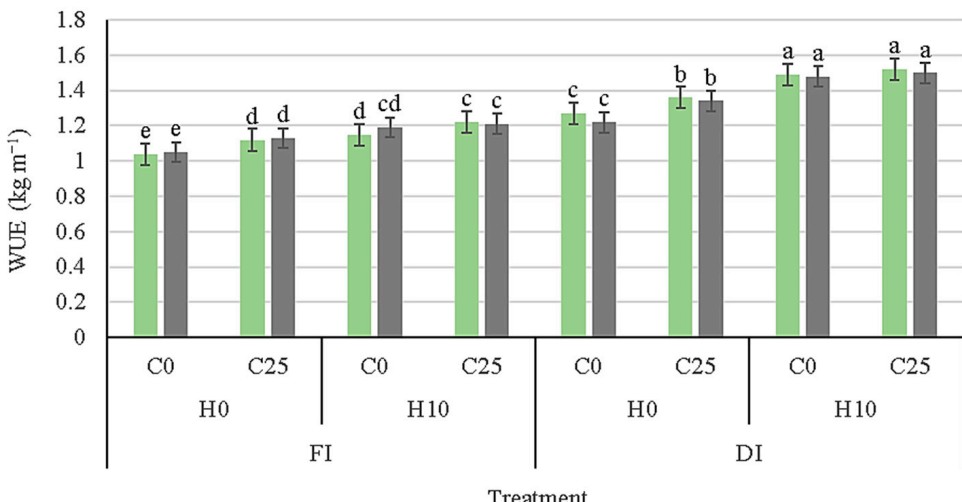

**Figure 4.** Water use efficiency (WUE) of faba bean as influenced by humic acid plus cytokinin treatments under irrigation regimes in the 2019/20 and 2020/21 seasons. Each value indicates the mean ± standard error (*n* = 3). Mean values in each bar followed by the same letter are not significantly different according to the Duncan test ($p \leq 0.05$). FI, full irrigation; DI, deficit in irrigation (80% of crop evapotranspiration); $H_0$ and $H_{10}$: without and with the application of 10 kg ha$^{-1}$ of humic acid, respectively. $C_0$ and $C_{25}$: without and with 25 mg L$^{-1}$ of cytokinin, respectively.

### 3.6. Chemometric Methods

Agglomerative hierarchical clustering (AHC) and principal component analysis (PCA) were utilized to present a collective understanding of the obtained data. With AHC (Figure 5), samples were clustered into two groups based on their dissimilarities. The first cluster contained the untreated FI and DI samples together with DI × $H_0$ plus $C_{25}$. The rest of the samples were grouped into the second cluster.

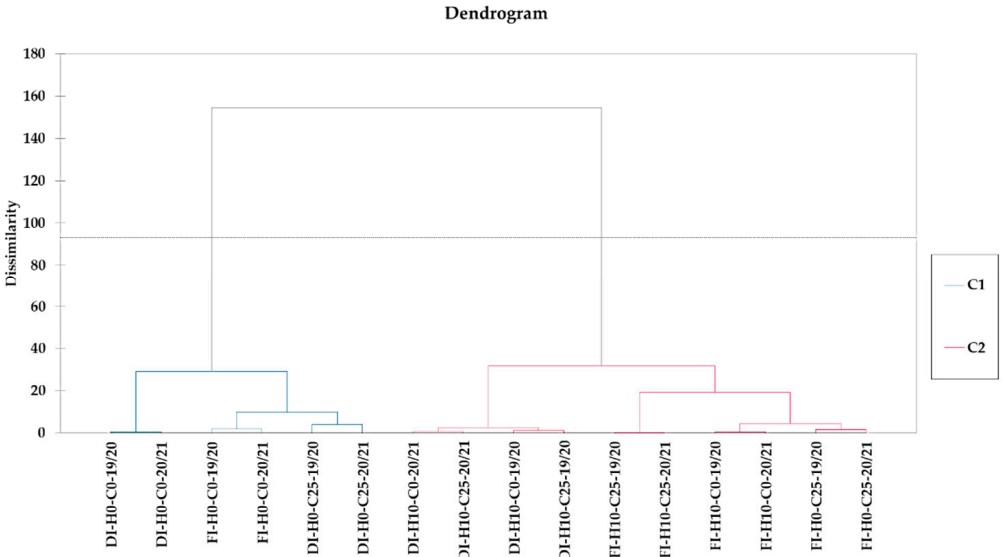

**Figure 5.** ACH clustering of the DI and FI samples.

According to the PCA biplot (Figure 6), two principal components can explain 87.65% of the variation (76.1% and 11.5 attributed to F1 and F2, respectively). Thereby, F1 can differentiate between the two irrigation systems regardless of their treatment method, except for DI × $H_0$ plus $C_0$ in the second season grouped with the FI samples. On the other hand, F2 can differentiate between the untreated and treated samples with humic acid or cytokinin regardless of their irrigation system, except for DI × $H_0$ plus $C_{25}$ in both seasons. Further, two major regions can be observed, marked in light purple and light green, indicating the samples that were clustered in the AHC test.

**Biplot (axes F1 and F2: 87.65 %)**

**Figure 6.** PCA biplot of the DI and FI samples.

It was clear that the DI samples treated with $H_{10}$ plus $C_0$ or $H_{10}$ plus $C_{25}$ in both seasons were correlated with the performance index, seed yield, WUE, TSS, Fv/Fm, SPAD, proline, and CAT as they were placed in the same quadrant (+F1/+F2). On the other hand, the FI samples treated with humic acid and/or cytokinin were correlated with the leaves, pods, and number of branches plant$^{-1}$, leaf area, the weight of 100 seeds, plant height, dry matter, the membrane stability index, and the relative water content as they were placed in the same quadrant (+F1/−F2).

## 4. Discussion

Plants exposed to water deficit exhibited changes in physio-biochemical status [55] and nutrient content [56], hence a reduction in growth and yield potential [57]. However, combined applications of humic acid and cytokinin mitigated the negative effects of drought, as evidenced in this research through the improvements in growth and physiology of faba bean. Plants under drought stress commonly close the stomata to reduce loss of water via transpiration [58]. However, stomatal closure led to a reduction in $CO_2$ inflow. In contrast, well-watered plants ensured $CO_2$ delivery through the stomatal apparatus [58]. Environmental stresses, specifically drought, adversely influenced plant pigments [59,60], particularly chlorophyll b [61]. Drought caused a decline in chlorophyll pigments and also accounted for reduced photosynthesis [62]. As a plant response to drought stress, plants develop a degree of drought tolerance through modulating gene functions that increase antioxidant defensive actions while reducing plant growth [63]. The reducing in soil moisture under drought stress adversely affected plant pigments and photosynthetic reactions, causing significant declines in crop growth and yield [64,65]. Accordingly, our findings revealed that supplying faba bean with low water (DI) without the exogenous application of humic acid plus cytokinin caused a reduction in SPAD, Fv/Fm, the relative water content, and the membrane stability index, hence reducing growth and yield.

To counteract the harms of drought, several actions should be adopted. In this respect, the defensive mechanisms of plants need to be exogenously equipped through specific compound applications. Amending the nutritional status of agricultural lands in favor of plant growth is crucial to increasing crop productivity [3]. In this context, soil structure, microorganism growth, plant growth and yield attributes were increased with humic acid application [66]. The uptake of several macro- and micronutrients was increased with humic acid supply [67,68]. By accelerating the rate of nutrient uptake, humic acid resulted in increases in plant growth, chlorophyll and protein content [69] and the photosynthetic rate [70]. Since humic acid increases micro- and macro-elements, activates enzyme, protein, sugar and vitamin synthesis, and alters the permeability of cell membranes [71–73], in addition to its high chelating potential [72], it increases crop yield. Accordingly, humic materials had a significant impact on plant growth and productivity under both normal and stress conditions [17,18].

The exogenous application of cytokinins during stress resulted in improvements in the membrane and chlorophyll stability indices, photosynthetic pigments, leaf relative water and soluble sugar content [74,75]. A range of processes related to plant growth and development, i.e., cell division, nutrient mobilization, tissue differentiation, the production of anthocyanin and retarding senescence, are influenced by cytokinins [30,76]. Additionally, cytokinins are significant to nitrogen and sulfur complements [77], causing inhibition of nitrate and sulphate uptake by plant roots [78,79]. Cytokinin increased the activity of superoxide dismutase, ascorbate peroxidase and catalase as well as ROS scavenging, and protected the cell membrane under abiotic stress [75]. The high concentration of cytokinin during osmotic stress resulted in several benefits such as a reduction in abscisic acid effects [80–85], a change in nutrient balance [86], and improvement in photosynthetic efficiency [87,88], hence leaf senescence was delayed [89]. The antagonistic work between abscisic acid and cytokinin led to the dominance of cytokinin, mediating the adverse effect of drought while regulating the developmental mechanisms in plants [90,91]. Recently, it has been documented that cytokinin is effective in alleviating stress through maintaining ion

balance [92]. Furthermore, at a molecular level, cytokinin improves photosynthesis under drought by adjusting the activity of proteins related to stomatal conductance, chlorophyll content and activation of rubisco [93].

Accordingly, our research work has provided insight into favorable changes by humic acid plus cytokinin for plants under drought. In this regard, providing drought-stressed faba bean plants with humic acid plus cytokinin maintained the appropriate water status, cell membrane stability, photosynthetic pigments and capacity (SPAD and Fv/Fm), as well as inducing osmo-protectants, particularly proline, catalase activity and nutrient absorption, as shown in Figure 7. Thus, humic acid–cytokinin-treated plants showed higher growth and yield in addition to nutrient content than non-treated plants.

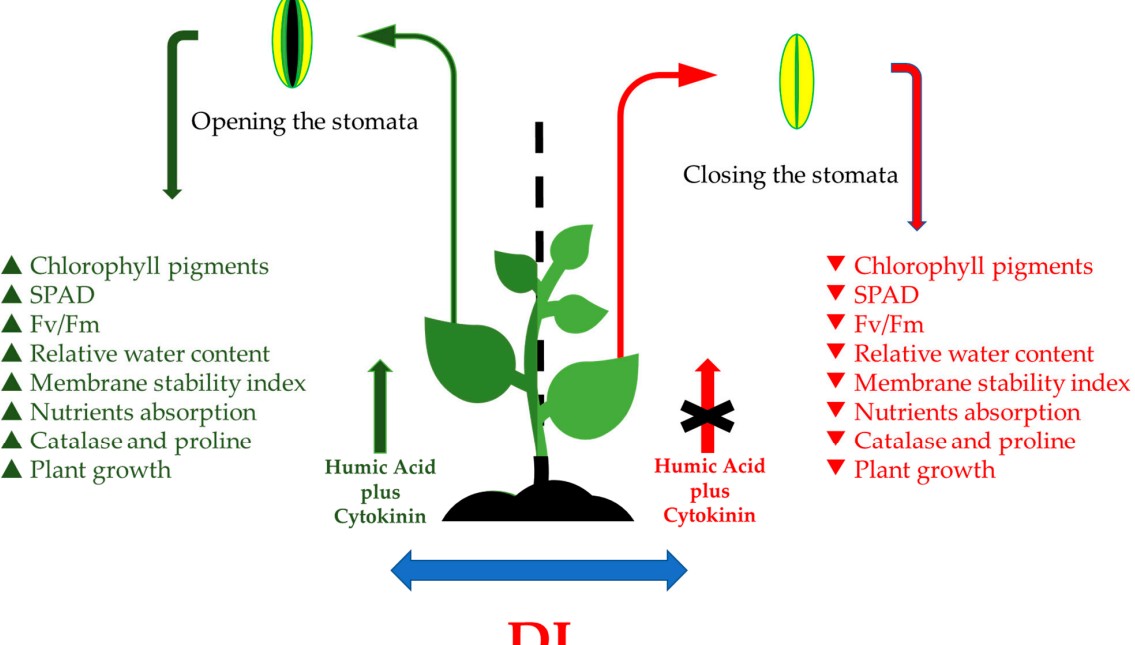

**Figure 7.** Illustration of the changes in physio-biochemical parameters in faba bean plants under deficit irrigation (DI) due to application of humic acid plus cytokinin for enhancing drought tolerance.

## 5. Conclusions

The use of a water-deficit strategy in crop irrigation, especially in arid and semi-arid regions, is a dire need for rationalizing the use of agricultural water. However, reduced water supply is associated with drought damages, which affect crop yield and quality. This research proved the complementary role of humic acid and cytokinin in ameliorating drought impacts by conserving water, nutrient balance and the photosynthesis apparatus of faba bean. Thus, the application of humic acid (10 kg ha$^{-1}$) and cytokinin (25 mg L$^{-1}$) is advisable for faba bean production, especially under drought stress conditions.

**Author Contributions:** Conceptualization, K.M.A.R., H.S.E.-B., T.A.A.E.-M., H.S.S., H.H.A.-O. and M.A.A.M.; methodology, K.M.A.R., H.S.E.-B., T.A.A.E.-M., H.S.S., H.H.A.-O. and M.A.A.M.; software, T.A.A.E.-M., H.S.S. and M.A.A.M.; validation, K.M.A.R., H.S.E.-B., T.A.A.E.-M., H.S.S., H.H.A.-O. and M.A.A.M.; formal analysis, K.M.A.R., H.S.E.-B., T.A.A.E.-M., H.S.S., H.H.A.-O. and M.A.A.M.; investigation, K.M.A.R., H.S.E.-B. and H.S.S.; resources, T.A.A.E.-M., H.S.S. and H.H.A.-O.; data curation, T.A.A.E.-M., H.S.S. and M.A.A.M.; writing—original draft preparation K.M.A.R., H.S.E.-B., T.A.A.E.-M., H.S.S., H.H.A.-O. and M.A.A.M.; writing—review and editing, K.M.A.R., H.S.E.-B., T.A.A.E.-M., H.S.S., H.H.A.-O. and M.A.A.M.; visualization, K.M.A.R., H.S.E.-B., T.A.A.E.-M., H.S.S., H.H.A.-O. and M.A.A.M.; supervision, K.M.A.R., H.S.E.-B. and H.S.S.; project administration, K.M.A.R., H.S.E.-B. and H.S.S.; funding acquisition, K.M.A.R., H.S.E.-B. and H.H.A.-O. All authors have read and agreed to the published version of the manuscript.

**Funding:** This research work was supported and funded by the Deputyship for Research and Innovation, Ministry of Education in Saudi Arabia (Project number INSTR001).

**Institutional Review Board Statement:** Not applicable.

**Informed Consent Statement:** Not applicable.

**Data Availability Statement:** All data are available within this manuscript.

**Acknowledgments:** The authors extend their appreciation to the Deputyship for Research and Innovation, Ministry of Education in Saudi Arabia for funding this research.

**Conflicts of Interest:** The authors declare no conflict of interest.

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
