# Peer review of "The Changes in Various Physio-Biochemical Parameters and Yield Traits of Faba Bean Due to Humic Acid Plus 6-Benzylaminopurine Application under Deficit Irrigation"

_agronomy, doi:10.3390/agronomy13051227_

Round 1
Reviewer 1 Report
The current research aimed to assess the integrated effect of humic acid and cytokinin on faba bean under water deficit. Authors concluded that the combined application of humic and plus cytokinin achieved physiological and nutrient homeostasis by adjusting the biochemical compounds in faba bean under deficit water. The present version requires major revisions, mainly introduction, results and discussion, and references.
First of all, title is too long and wordy. It should be shortened.
Divide the introduction into 3 paragraphs to clearly understand the need and background. In introduction and discussion, you can add some recent publications, such as doi: 10.1002/tpg2.20279.
Overall, the results are presented in a shallow manner. Authors should also provide numerical data for significant results, such as folds/times etc., to understand the positive impact of HA.
The discussion is too short and needs to be significantly improved.
To further comprehend the HA and CK-mediated drought tolerance process, authors might provide a new mechanistic model figure.
Too many self-citations have been noticed in the reference list. While the introduction and discussion are too short, the authors added a long list of references, including their own papers. The number of references should be decreased by half.
Numerous writing mistakes can be found. Please make sure the document is legible and free of grammatical errors by carefully checking the language, syntax, and grammar.
Author Response
April …., 2023
Subject: Submission of Revised manuscript entitled ”The Changes in Various Physio-Biochemical Parameters and Yield Traits of Faba Bean Owing to Humic Acid plus 6-Bеnzylаminоpurinе Application under Deficit Irrigation (agronomy-2343851)" for publication in Agronomy
Dear Prof. Anita Pizurica
Section Managing Editor, MDPI Belgrade
With our best greetings
It is pleasure to us to submit our revised article entitled ”The Changes in Various Physio-Biochemical Parameters and Yield Traits of Faba Bean Owing to Humic Acid plus 6-Bеnzylаminоpurinе Application under Deficit Irrigation” for publication in Agronomy. We state that the work is original. It has not been published and not being considered for publication elsewhere in its final form, in printed or in electronic format.
We responded to all editor and Reviewer comments and addressed all required corrections (See Author Response below)
We hope this revised article takes your acceptance.
Thanks.
Yours’ sincerely
Corresponding author
Authors
Author Response
Dear Respected Editor and Reviewers,
Thank you so much for reviewing and editing our manuscript. We thoroughly revised and modified the manuscript guided with the useful editorial and reviewer’s comments and suggestions. Please find the revised and modified manuscript, in addition to the response letter replying to point-by-point received comments from editor and reviewers.
Comments |
Author Response |
Notes |
Reviewer #1, all required correction Highlighted in yellow |
|
|
The current research aimed to assess the integrated effect of humic acid and cytokinin on faba bean under water deficit. Authors concluded that the combined application of humic and plus cytokinin achieved physiological and nutrient homeostasis by adjusting the biochemical compounds in faba bean under deficit water. The present version requires major revisions, mainly introduction, results and discussion, and references. |
All suggested corrections have been made |
See the whole manuscript |
First of all, title is too long and wordy. It should be shortened |
Revised and shortened |
See the title |
Divide the introduction into 3 paragraphs to clearly understand the need and background. In introduction and discussion, you can add some recent publications, such as doi: 10.1002/tpg2.20279. |
Introduction section improved and supported by recent citations |
See the Introduction section, specifically the highlighted lines |
Overall, the results are presented in a shallow manner. Authors should also provide numerical data for significant results, such as folds/times etc., to understand the positive impact of HA. |
Results revised and presented in good manner as suggested. We supported the presentation with increases percentages for the significant values. Please note that we presented the result data focusing on the positive effects of humic acid and cytokinin combination of faba bean traits while not increase the length of the manuscript. |
See the Results section, specifically the highlighted lines |
The discussion is too short and needs to be significantly improved. |
The discussion improved |
See the Discussion section, specifically the highlighted lines |
To further comprehend the HA and CK-mediated drought tolerance process, authors might provide a new mechanistic model figure. |
Done |
See the Discussion section, specifically the last highlighted paragraph |
Too many self-citations have been noticed in the reference list. While the introduction and discussion are too short, the authors added a long list of references, including their own papers. The number of references should be decreased by half. |
The suggested citations are so beneficial, and we believe that this will improve and contribute to the article. However, we reduced the references as possible as we can. Please note that, since we assessed various parameters as described in Materials and Methods section, for these parameters, about 20% of the total references (17of 87 citations) were cited for this purpose. These references are mandatory. Also, we reduced the self-citations by half |
See the references list and citations |
Numerous writing mistakes can be found. Please make sure the document is legible and free of grammatical errors by carefully checking the language, syntax, and grammar. |
The language has been revised and improved throughout the manuscript |
See the manuscript overall |

Reviewer 2 Report
Dear Authors,
I congratulate you on the draft article, but I have a few suggestions for changes:
Keywords must be written with a small initial letter
It would be better to separate the literary references a little, not to refer to them in such a large proportion. Please refer to each clause where possible.
The Introduction chapter is somewhat short compared to the length of the article. I think that this part can be supplemented and fulfilled, because the topic justifies that you can read more about the background and topic of the research.
It is worth supplementing the article with a Conclusions chapter, considering that both the article and the Discussion are large in length.
Author Response
Dear Respected Editor and Reviewers,
Thank you so much for reviewing and editing our manuscript. We thoroughly revised and modified the manuscript guided with the useful editorial and reviewer’s comments and suggestions. Please find the revised and modified manuscript, in addition to the response letter replying to point-by-point received comments from editor and reviewers.
Reviewer #2, all required correction Highlighted in blue |
|
|
I congratulate you on the draft article, but I have a few suggestions for changes: |
Thank you so much |
|
Keywords must be written with a small initial letter |
Done |
See the keywords |
It would be better to separate the literary references a little, not to refer to them in such a large proportion. Please refer to each clause where possible. |
OK, done |
See the introduction and discussion sections |
The Introduction chapter is somewhat short compared to the length of the article. I think that this part can be supplemented and fulfilled, because the topic justifies that you can read more about the background and topic of the research. |
Introduction section improved and supported by recent citations |
See the Introduction section, specifically the highlighted lines |
It is worth supplementing the article with a Conclusions chapter, considering that both the article and the Discussion are large in length. |
Done, conclusion added |
See the conclusion section |

Round 2
Reviewer 1 Report
Though authors have made the suggested corrections. But one of my suggestions has not been incorporated. To further comprehend the HA and CK-mediated drought tolerance process, authors might provide a new mechanistic model figure.
The suggested model can be cited in the last paragraph as shown in author response. It will help readers understand the mechanism.
Author Response

(The authors gave the same response as above.)

Author Response
Dear Respected Editor and Reviewers,
Thank you so much for reviewing and editing our manuscript. We thoroughly revised and modified the manuscript guided with the useful editorial and reviewer’s comments and suggestions. Please find the revised and modified manuscript, in addition to the response letter replying to point-by-point received comments from editor and reviewers.
Comments |
Author Response |
Notes |
Reviewer #1 |
|
|
Though authors have made the suggested corrections. But one of my suggestions has not been incorporated. To further comprehend the HA and CK-mediated drought tolerance process, authors might provide a new mechanistic model figure. The suggested model can be cited in the last paragraph as shown in author response. It will help readers understand the mechanism. |
All suggested corrections have been made |
Figure 7. Illustration of the changes in physio-biochemical parameters in faba plants under low water supply in response to humic acid plus cytokinin meditates drought tolerance.
|
